# Quantification of the Chemical Chaperone 4-Phenylbutyric Acid (4-PBA) in Cell Culture Media via LC-HRMS: Applications in Fields of Neurodegeneration and Cancer

**DOI:** 10.3390/ph16020298

**Published:** 2023-02-14

**Authors:** Salvatore Villani, Giulia Dematteis, Laura Tapella, Mara Gagliardi, Dmitry Lim, Marco Corazzari, Silvio Aprile, Erika Del Grosso

**Affiliations:** 1Department of Pharmaceutical Sciences, University of Piemonte Orientale, Largo Donegani 2, 28100 Novara, Italy; 2Department of Health Science (DSS), Center for Translational Research on Autoimmune and Allergic Disease (CAAD) & Interdisciplinary Research Center of Autoimmune Diseases (IRCAD), University of Piemonte Orientale, 28100 Novara, Italy

**Keywords:** 4-phenylbutyric acid, LC-HRMS, method validation, in vitro treatment, Alzheimer’s disease astrocyte, melanoma, ER stress, cellular absorption, adsorption

## Abstract

In recent years, 4-phenylbutyric acid (4-PBA), an FDA-approved drug, has increasingly been used as a nonspecific chemical chaperone in vitro and in vitro, but its pharmacodynamics is still not clear. In this context, we developed and validated a Liquid Chromatography–High Resolution Mass Spectrometry (LC-HRMS) method to quantify 4-PBA in NeuroBasal-A and Dulbecco’s Modified Eagle widely used cell culture media. Samples were injected on a Luna^®^ 3 µm PFP(2) 100 Å (100 × 2.0 mm) column maintained at 40 °C. Water and methanol both with 0.1% formic acid served as mobile phases in a step gradient mode. The mass acquisition was performed by selected ion monitoring (SIM) in negative mode for a total run time of 10.5 min at a flow rate of 0.300 mL/min. The analogue 4-(4-Nitrophenyl)-Butyric Acid served as internal standard. Validation parameters were verified according to FDA and EMA guidelines. The quantification ranges from 0.38–24 µM. Inter and intraday RSDs (Relative Standard Deviations) were within 15%. The developed LC-HRMS method allowed the estimation of 4-PBA absorption and adsorption kinetics in vitro in two experimental systems: (i) 4-PBA improvement of protein synthesis in an Alzheimer’s disease astrocytic cell model; and (ii) 4-PBA reduction of endoplasmic reticulum stress in thapsigargin-treated melanoma cell lines.

## 1. Introduction

Chemical chaperones are small molecules, usually osmolytes with a molecular weight of <900 Da, which nonspecifically interact with proteins and generate a solvophobic force capable of preventing denaturation and promoting protein trafficking improvements [1,2]. Nowadays some chemical chaperones, such as glycerol, valienamine derivative (N-octyl-4-epi-β-valienamine), and 4-phenylbutyric acid (4-PBA) are used for Lysosomal Storage Disorders (LSDs) treatment [3,4,5]. Sodium phenylbutyrate (4-PBA sodium salt) is a prodrug approved by the FDA and indicated as adjunctive therapy in urea cycle disorders. 4-PBA is rapidly metabolized to phenylacetate, an active metabolite, that conjugates with glutamine via acetylation to form phenylacetylglutamine, which is finally excreted by kidneys. In this way, an alternative mechanism for waste nitrogen excretion is provided [5]. However, 4-PBA is known to promote the reduction of endoplasmic reticulum (ER) stress both in vivo and in vitro by acting as a chemical chaperone [6,7,8]. In the past decades, this activity has been a subject of interest in several fields from neurobiology to oncology, although its pharmacodynamic mechanism is not completely understood [2,6,9,10].

Many in vitro studies are arising with the aim of identifying the mechanism of action of 4-PBA, and to describe the biomodulation driven by this drug, or using 4-PBA as an ER stress and protein synthesis modulator [2,6,11,12,13]. Different biological readouts have been used to describe the 4-PBA effects, such as analysis of the ER stress modulation through quantitative real-time PCR (qPCR) or Western blot (WB), or protein synthesis induction via the Surface Sensing of Translation (SUnSET) method [7,12,13,14,15]. In this context, a method to assess 4-PBA absorption in different cell types would be useful to better define the treatment concentration, and assess the implication of co-treatment with other drugs.

Recently, Tapella and collaborators analyzed the cellular secretome and demonstrated that 4-PBA treatment of an immortalized astrocytic cell line (3Tg-iAstro cells) from 3xTg-AD mice (Triple-Transgenic model (3xTg-AD) harboring PS1(M146V), APP(Swe), and tau(P301L) transgenes), a popular model of Alzheimer’s disease (AD), rescued the presence of proteins responsible for extracellular matrix formation [2]. In light of this study and the increasing interest in cell culture media, including whole secretome, extracellular vesicles, and disease treatment, it might be of paramount importance, for the correct design of experiments, to know the stability and absorption of 4-PBA [16,17,18].

To the best of our knowledge, the determination and quantification of 4-PBA in cell culture media have not been particularly dealt with in bioanalytical research. In the relevant literature, several studies have reported the effect and concentration of 4-PBA and its metabolites in biological matrices, such as plasma and urine, using liquid and gas chromatography methods coupled with diode array detectors (DAD) or mass spectrometry (MS) [19,20,21,22]. However, none of these methods was developed to quantify 4-PBA in cell culture media, which are known to be rich of interferents (amino acids, vitamins, growth factors, proteins, salts, etc.) potentially causing ion suppression and severe matrix effects [23].

In the last decades, the development of LC-HRMS methods to quantify analytes in vitro have been acquiring much more interest into the pharmaceutical field. In this context, in vitro assessment of 4-PBA extracellular concentration could be a new strategy to underpin many pharmacological approaches, widely spread in the literature which have been comparing its biological effects to the surrounding cellular microenvironment.

In this investigation, we describe the development and validation of a new LC-HRMS method to quantify 4-PBA, but also to monitor possible combined treatment, in order to investigate its in vitro behavior and effects. The validation was successfully performed in Dulbecco’s Modified Eagle Medium (DMEM), one of the most common medium for cell culture and Neurobasal A medium (NBA), a specific medium for CNS (Central Nervous System) cells’ support. We correlated 4-PBA concentration with important biological effects including: protein synthesis recovery in Immortalized AD astrocytes by Western blot, and gene expression of ER stress markers (Activating Transcription Factors, ATF4 and ATF6) in melanoma cell lines by qPCR. Moreover, we also analyzed the adsorption kinetics of 4-PBA in functionalized cell culture plates.

## 2. Results and Discussion

### 2.1. Development and Optimization of LC-HRMS Conditions and Sample Preparation

Firstly, different solid-phase extraction (SPE) procedures and cartridges were tested, in order to increase the 4-PBA detection sensitivity by cleaning up matrix interferences. Some examples of the tested cartridges are Oasis MAX 3cc (60 mg), Phenomenex Strata SAX (55 μm, 75 A) 100 mg/3 mL, and Waters Sep-Pak Vac 1cc (100 mg) C18 cartridges. Although sample preparation by SPE provided good recovery of the analyte, the procedure suffered a lack of reproducibility. Even though we had reached some improvements in SPE, they were not enough for procedure validation purposes. Hence, the protein precipitation approach with methanol was performed, because it resulted in being more reproducible and saved time. Different dilution ratios were tested, i.e., 1:1, 1:2 and 1:3. The first two ratios failed to precipitate the proteins efficiently, providing cloudy supernatants; conversely, the 1:3 dilution allowed for clear supernatants to be obtained. A solvent evaporation step was executed to cancel the previous dilution, allowing to reach a low LOD value and clear samples. However, a C18 SecurityGuard precolumn cartridge (4 × 2.0 mm—Phenomenex, Torrance, CA, USA) was used to further protect the column.

Labelled internal standards would be preferred for their strict correlation with analyte and being distinguishable by MS. However, we settled on NPBA, a less expensive exogenous compound than the 4-PBA deuterated analogue (d-4-PBA). NPBA is chemically related to 4-PBA with a nitro group in position 4 on benzene ring, which adds a difference of 45 *m*/*z* detectable by MS. Chemical structures and chromatograms of both analyte and IS are shown in Figure 1, whereas MS/MS (tandem mass spectrometry) spectra used for their structural confirmation are reported in the Appendix A.

Another critical point was the column choice due to 4-PBA peak tailing, which probably interacts with free silanol groups at the silica surface. In the first instance, a Kinetex 2.6u C18 100 Å (100 × 2.10 mm) column was tested, but it displayed a significant peak tailing. Since 4-PBA and NPBA have both a phenyl ring, we tested two different phenyl stationary phases to aim for a better separation. The two columns were a Kinetex 5u Biphenyl 100 Å (100 × 2.10 mm) and a Luna^®^ 3 µm PFP(2) 100 Å (100 × 2.0 mm). The latter gave the better resolution and good retention even when operating at the high percentage of organic modifier. The analytical runs were conducted at a constant temperature of 40 °C to reduce variability plus slightly improve the peak symmetry and analysis speed. Acetonitrile and methanol were tested as organic modifiers to obtain narrow peaks. Acetonitrile, which usually provides sharpening of the chromatographic peaks, did not make any improvement in this case. Methanol gave similar results but longer retention time, which provides analytes retention time out of the dead volume. Several gradient and isocratic modes were tested to achieve symmetric peaks plus acceptable retention time. Finally, we adopted the step gradient because it gave fairly resolved and symmetric peaks. Moreover, the isocratic step operating at 100% of organic modifier suited well for both column washing and monitoring lipophilic compounds. Hence, we were able to semi-quantify thapsigargin, even if it was not our main aim, we confirmed its operating concentrations (Appendix A). Furthermore, its quantification may later be useful, if it will be considered appropriate for supporting other biological data.

Method specificity was verified during the development phase, to establish which MS monitoring mode, between PRM and SIM, was more sensitive and specific for the detection of 4-PBA. In this way, we could choose the best compromise between sensitivity and specificity by considering which mode gave the highest signal-to-noise (S/N) value at low concentration levels and minimal interference. The comparison between SIM and PRM detections of 4-PBA is illustrated in Figure 1. The S/N obtained at the MQC was 324 using SIM and 76 for PRM. Hence at the end, we settled on SIM data acquisition.

#### Validation and Chromatography/Mass Spectrometry Analyses

In the current research, we proposed a quantitative LC-HRMS method to evaluate the cellular response according to the micro-environmental concentration of 4-PBA. To the best of our knowledge, our method represents the first validated system for the assessment of 4-PBA in NBA (NeuroBasal-A medium) and DMEM (Dulbecco’s Modified Eagle Medium) matrices.

Linear relationships between the nominal analyte concentration and its instrumental response were confirmed in both matrices from 0.38 µM to 24 µM. Calibration curves have a determination coefficient (r^2^) > 0.99 and intercept nearly to zero or origin. LLOD for 4-PBA was identified at 0.19 µM. Moreover, our LLOQ of 0.38 µM is less than other LC-HRMS procedures reported in the literature enabling the quantification of lower concentrations [19,20].

Precision, accuracy (AC), matrix effect (ME), and recovery data are listed in Table 1 and Table 2, respectively for NBA and DMEM. Intra and inter-day precisions (%RSDs) were within ±15% of nominal concentrations even at LLOQ concentration. Intra and between-run accuracies respected the acceptance criteria ranging from a minimum of −10.31% to 13.89%.

The matrices employed in this work were cell culture media with a specific and reproducible composition. Hence, matrix interferences were negligible; in addition, mass detection provided excellent selectivity.

Regarding the matrix effect, the analyzed matrices had a different impact. NBA caused a slight response enhancement (mean ME = 103.5%), whereas DMEM tended to suppress (mean ME = 70.7%). FDA and ICH guidelines suggest that recovery evaluation is critical for methods which employ sample extraction [24,25,26,27]. In this work, a protein precipitation with methanol followed by partial evaporation was employed. The procedure to quantify the recovery, as described above, can be considered reasonable, because stability after solvent evaporation was confirmed. Hence, recovery samples (QCs) were prepared by adding blank matrices to completely evaporate residuals including both 4-PBA and IS, as described in Section 3.4.2. Mean recovery at each QC level was >90% (range 92.5–114.2%) and reproducible with %RSDs < 10%. 4-PBA and the IS (NPBA) were stable; no response alteration was found after bench-top, freeze–thaw, partial or complete evaporation and rack stability assays. For these assays, precision and accuracy (AC) are shown in Table 3 and Table 4. Finally, all parameters required for validation fall within the limits of EMA, FDA, and ICH guidelines.

### 2.2. 4-PBA Quantification in Astrocyte-Conditioned Neurobasal A Medium

As published previously, 4-PBA treatment is able to rescue the defect of protein synthesis described in 3Tg-iAstro cells [2,26]. Here, we investigated if 4-PBA that remains in the media after 48 h of treatment of WT-iAstro (ACM + 4-PBA 48 h) is still able to induce the increase of protein synthesis in both WT and Tg-iAstro. With this aim, we designed an experiment consisting of two steps, as shown in Figure 2A. In the first step, WT-iAstro were plated and treated or not with 4-PBA (3 µM, 48 h). Astrocyte conditioned medium from iAstro that was not treated (ACM) was collected and stored. Part of these samples were used for the LC-HRMS assessment of 4-PBA concentration in the ACM, while another part was used in the second step of the experiment in which WT-iAstro and 3Tg-iAstro cells were treated with (i) ACM collected from WT-iAstro, treated with vehicle (ACM); (ii) ACM collected from WT-iAstro, treated with vehicle to which 3 µM of fresh 4-PBA was added (ACM + Fresh 4-PBA); (iii) ACM collected from WT-iAstro treated with 4-PBA (3 µM) for 48 h (ACM + 4-PBA 48 h). After 48 h, cells were pulse-labelled with puromycin and protein synthesis rate was assessed via WB (Figure 2B). ACM + 4-PBA 48 h, in which a concentration of 4-PBA 0.38 ± 0.02 µM was present in the medium, was still able to increase protein synthesis in WT-iAstro but had no effect on 3Tg-iAstro cells, suggesting that higher 4-PBA concentrations are necessary to rescue protein synthesis in 3Tg-iAstro (Figure 2C). It should also be noted that the 4-PBA concentration in the medium 48 h after treatment, which is about 10% of the initial 4-PBA concentration, was not negligible and was still able to induce a significant effect in WT-iAstro. These results emphasize the importance of monitoring the concentration of 4-PBA in culture media during treatment (Figure 2D).

### 2.3. 4-PBA Quantification in Melanoma Cell-Lines Conditioned Dulbecco’s Modified Eagle Medium

We previously described the ability of 4-PBA to inhibit the oncogenic BRAF-induced chronic ER stress, thus enhancing the sensitivity of melanoma cells to TG-induced cell death [15]. Here, we evaluated whether the absorption of 4-PBA by cells might be affected by the concentration of TG. To this aim, both CHL-1 and A375 cells were exposed to 5 µM 4-PBA in the absence or presence of TG ranging from 1–10 µg/mL, and the expression of well-known ER stress markers was evaluated by qPCR analysis. DMEM alone and DMEM+4-PBA (5 µM) were used as negative controls. As shown in Figure 3A,B, we observed a (TG) dose-dependent enhanced expression of both the ER stress markers ATF4 and ATF6 in both cell lines, which was almost completely abrogated by the presence of 4-PBA independently from TG concentration.

These data clearly indicate that 5 µM 4-PBA was able to inhibit the TG-induced ER stress, at least in the range of concentrations used in these experiments (1–10 µg/mL). Moreover, measuring the amount of 4-PBA in the cell-conditioned medium after 24 h, we observed that cell absorption of the chemical chaperone is not influenced by the presence or absence of TG, nor by the concentration of TG (Figure 3A,B right panels). Hence 4-PBA intracellular concentration of about 1.00 µM was enough to normalize TG-induced ER stress in all experimental conditions (Figure 3A,B).

### 2.4. Adsorption of 4-PBA

During the sample studies, we found out some 4-PBA apparent loss between initial time (t0) and incubation final time (24 h or 48 h) although without cells. Firstly, we hypothesized that culture dishes, commonly employed for cell adhesion, could promote 4-PBA direct adsorption. Chemically, 4-PBA is an acidic drug with a pKa of 4.78 completely dissociated at an experimental pH of 7.4 [28]. Consequently, anionic 4-PBA is repelled by negatively charged functional groups onto the surface of pre-treated dishes. Hence, plasma-treated polystyrene plastics should not significantly adsorb acidic drugs, as reported in the literature [29]. To assess if container materials adsorb 4-PBA, we compared its concentration at the outset (t0) and after 48 h (t48) on Pyrex glass petri (G), non-treated polystyrene petri (NTP), and plasma-treated polystyrene dishes (PTP), as reported in the Section 3.7.4. The plotted adsorption curves showed a similar trend with r^2^ values of 0.8265 in the case of NBA (Figure 4A) and 0.8792 in DMEM medium (Figure 4B).

Their estimated adsorption loss referred to the 4-PBA initial concentration at t0 (CTRL) were 60.7% and 40.9% in NBA and DMEM, respectively. Adsorption kinetics are reproducible and reach a steady state level after only one hour of exposure (Figure 4A,B). Figure 4C showed a similar loss of 4-PBA in all considered materials, that might be hypothetically caused by Albumin/4-PBA bond. Albumin (ALB) is a single chain and non-glycosylated protein, well known as a carrier of both exogenous and endogenous substances into the bloodstream [30]. ALB is highly affine to anionic substances, such as acidic xenobiotics and fatty acids. Consequently, ALB is included in culture medium by B27 or FBS supplement, which might reasonably interact with 4-PBA. Furthermore, Albumin tends to adhere on anionic-charged PTP, so ALB/4-PBA complex may be well detained resulting in the indirect adsorption of 4-PBA, which might explain the reduction of 4-PBA free concentration during incubation [30,31].

## 3. Materials and Methods

### 3.1. Reagents

4-phenylbutyric acid (4-PBA) was supplied from Santa Cruz Biotechnology (Dallas, TX, USA, Cat. sc-232961); 4-(4-nitrophenyl)butyric acid (NPBA, Sigma-Aldrich, Milan, Italy; Cat. N20506, internal standard, IS), thapsigargin (Sigma-Aldrich; Cat. T9033), LC-HRMS grade formic acid, methanol, and water were purchased from Carlo Erba (Milan, Italy, Cat. 414831 and 412111). DMEM and NBA matrices were obtained from Sigma-Aldrich and Thermo Fisher Scientific (Waltham, MA, USA), respectively.

### 3.2. Preparation of Standard Solutions

4-PBA stock solution (SS) was prepared by dissolution in methanol to reach a concentration of 20 mM. Working solution (WS) of 4000 µM was obtained by diluting SS in H_2_O:MeOH 1:1 mixture. Separate stock and working solutions were used to prepare matrix-based samples, including calibrators and quality controls (QCs) samples. Seven calibrators were prepared at ranging concentrations from 0.38 to 24 µM including LLOQ (Lower Limit of Quantification, 0.38 µM), low (LQC = 0.75 µM), medium (MQC = 12 µM), and high (HQC = 18 µM) Quality Controls (QCs).

IS stock and its diluted solutions were prepared using methanol as solvent. IS stock 20 mM solution was diluted to reach the final concentration of 0.02 µM and used to precipitate matrix proteins during sample preparation. Blank and zero samples were made using free-analyte matrices. In the first case, IS was not added either. Matrix-based standards (calibrators, QCs, blank and zero samples) were freshly prepared.

### 3.3. Sample Preparation

Matrix-based standards and study samples were treated as follows: 100 µL aliquots of each sample were collected in a microcentrifuge tube, where 100 µL of IS solution 0.02 µM plus 200 µL of methanol were previously added. The diluted samples were vortexed for at least 1 min and centrifuged at 13,000 rpm for 5 min; 200 µL of supernatant was transferred in a new microtube and the solvent evaporated in a centrifugal evaporator at 1500 rpm, 60 °C for 35 min; 65 µL of residual supernatant was analyzed by LC-HRMS.

Blank samples were prepared following the same procedure except for the addition of the IS aliquot which was replaced by 100 µL of pure methanol.

### 3.4. Instrumentation and Chromatographic Conditions

#### 3.4.1. LC-HRMS Analyses

A Q-Exactive Plus UHMR Hybrid Quadrupole Orbitrap™ Mass Spectrometer equipped with a Vanquish™ Duo UHPLC system was employed (Waltham, MA, USA). The chromatographic separation was performed using a Luna^®^ 3 µm PFP(2) 100 Å (100 × 2.0 mm) column protected by a C18 SecurityGuard precolumn cartridge (4 × 2.0 mm—Phenomenex, Torrance, CA, USA) maintained at 40 °C. The eluant was made of 0.1% formic acid in water UHPLC grade (solvent A) and 0.1% formic acid in methanol UHPLC grade (solvent B) at the flow rate of 0.300 mL/min. A step gradient was set as follows: A-B (40:60, *v*/*v*) for 5.0 min, A-B (0:100, *v*/*v*) in 0.1 min, A-B (0:100, *v*/*v*) from 5.1 to 8.0 min, A-B (40:60, *v*/*v*) from 8.1 to 10.5 min, that is the total run-time. The injection volume was 5 µL. The negative ionization mode was performed in the following operating conditions: spray voltage, 2.80 kV; source current, 1.80 μA; capillary temperature, 300 °C; sheath gas flow (N_2_) 45.00 L/min; sweep gas (N_2_) flow 0.00 L/min; Aux gas (N_2_) flow rate 10.00 L/min heated to 300 °C. After preliminary full-scan acquisitions (scan range 50 to 750 *m*/*z*), the data were collected in selected ion monitoring (SIM) including: 4-PBA ([M-H]^−^: 163.07645 *m*/*z*), NPBA (IS) ([M-H]^−^: 208.06153 *m*/*z*) and thapsigargin (TG) ([M-H]^−^: 649.32295 *m*/*z*), which chromatograms are reported in Appendix A. MS spectra were acquired and processed using Xcalibur^®^ software (Thermo Fisher Scientific). A summary table of our LC-HRMS method is available in the Appendix A.

#### 3.4.2. LC-HRMS Method Validation

The method was validated according to the bioanalytical method validation guidelines provided by both FDA and EMA [24,25]. The ICH M10 guideline draft version was taken into account, as well [26]. The method validity was tested in terms of linearity, accuracy, precision, recovery, selectivity, matrix effect, carry-over, and stability. The details of a general bioanalytical validation can be found in the Appendix A.

##### Linearity

The calibration curve was plotted over seven concentration levels ranging from 0.38 to 24 µM, in the presence of the IS. Method validation experiments were conducted over three independent runs by using freshly prepared standard solutions as described in Section 3.2 and Section 3.3. The instrument response was calculated as the ratio between the analyte and IS area after integration by Xcalibur^®^ software. The relationship between 4-PBA/IS peak area ratios and analyte concentration was graphed by using 1/x weighted linear regression.

##### Accuracy and Precision

Within-run accuracy (AC) and precision were determined for each QC level by injecting the same sample five times. For between-run validation, each QC level was evaluated in three runs over three different days.

##### Matrix Effect and Recovery

The matrix effect was evaluated by comparing the response of the analyte spiked in the extracted matrix with the response of the analyte dissolved in pure methanol at low, medium, and high concentrations. Each level was analyzed in triplicate.

The recovery was assessed as follows: 4-PBA methanolic solutions at each QC level (low, medium, and high) including IS were prepared and the solvent completely evaporated by concentrator centrifuge operating at 60 °C for 35 min. Hence, blank matrix (65 µL) was added and the recovery samples were processed as reported in Section 3.3 except for the addition of IS.

##### Stability Assays and Carry-Over

Bench-top, freeze–thaw, long-term, autosampler stabilities were assayed by analyzing low, medium, and high QC in triplicate. Bench-top stability was evaluated by analyzing QC samples after storage at room temperature for 6 h, which is longer than the time usually needed for whole sample preparation (about 1 h and a half). Freeze–thaw stability assay was carried out over three freeze–thaw cycles from −15 °C to room temperature. Long-term and autosampler stabilities were estimated together by analyzing QC samples after storage at 15 °C for 24 and then 48 h. Stability after solvent evaporation was also assayed. Carry-over was measured by analyzing blank samples after the 24 µM calibrator, which is the ULOQ (Upper Limit of Quantification).

### 3.5. Cell Lines

#### 3.5.1. Immortalized Hippocampal Astrocytes from WT and 3xTg-AD Mice

Immortalized astrocytes (iAstro) from hippocampi of WT (wild type) and 3xTg-AD mice (WT- and 3Tg-iAstro cells) were generated in our laboratory via transduction using retrovirus expressing SV40 large T antigen. The procedural details are available in the following reference [27]. iAstro lines were maintained in complete culture media containing Dulbecco’s modified Eagle medium (DMEM; Sigma-Aldrich, Cat. D5671) supplemented with 10% fetal bovine serum (Gibco, Cat. 10270) (FBS), 2 mM L-glutamine (Sigma-Aldrich), and 1% penicillin/streptomycin solution (Sigma-Aldrich). Cells were passaged once a week and used for experiments between passages 12 and 20 from the establishment.

#### 3.5.2. Melanoma Models A375 and CHL-1 Cell Lines

Melanoma cell lines, CHL-1 (Cellosaurus Resource Identification Initiative, RRID: CVCL_1122) and A375 (Cellosaurus RRID: CVCL_0132) were a gift from Prof. P.E. Lovat (Newcastle University, Newcastle Upon Tyne, UK).

CHL-1 (BRAF wild type) and A375 (BRAFV600E) cell lines were cultured in DMEM (Sigma-Aldrich) supplemented with 10% fetal bovine serum (Sigma-Aldrich), 2 mM L-glutamine (Sigma-Aldrich), and 1% penicillin/streptomycin solution (Sigma-Aldrich) at 37 °C under 5% CO_2_.

### 3.6. Media Preparation and Cell Treatments

#### 3.6.1. Astrocytes Conditioned Medium (ACM) Preparation

For the preparation of ACM, 5 × 10^4^ WT-iAstro or 3Tg-iAstro cells were plated in a 6-well plate. After 24 h the media was changed with neurobasal A medium (Invitrogen, Cat. 10888022) supplemented with 2% B27 supplement (Invitrogen, Waltham, MA, USA, Cat. 17504044), 2 mg/mL glutamine, 10 U/mL penicillin, and 100 mg/mL streptomycin. 48 h later, the media were collected and centrifuged at 12,000 rpm for 10 min at 4 °C. ACM was stored at −80 °C.

#### 3.6.2. Treatment WT-iAstro and 3xTg-iAstro Cells with 4-Phenylbutyric Acid (4-PBA)

WT-iAstro and 3Tg-iAstro cells were plated, and after 24 h were treated with 3 µM 4-PBA (Santa Cruz Biotechnology, Dallas, Texas, USA, Cat. sc-232961), 48 h later, and the media were collected and processed as already described (Section 3.3).

#### 3.6.3. Treatment of CHL-1 and A375 Cells with 4-Phenylbutyric Acid (4-PBA)

Treatment involved 25 × 10^4^ cells/well being seeded in 6-well plates (Sarstedt, polystyrene wells), and cells were exposed to a range of concentrations of TG (1, 5, and 10 µg/mL) in the presence or absence of 5 µM 4-PBA. After 24 h, cell media were collected and centrifuged at 12,000 rpm for 10 min at 4 °C and stored at −80 °C.

### 3.7. Applications

#### 3.7.1. Astrocytes Experiments

##### Astrocytes Treatment with ACM and Protein Synthesis Assessment

Astrocyte treatment involved 2 × 10^4^ WT- and 3xTg-iAstro cells being plated in a 24 MW plate. After 24 h, the media were changed with the ACM. After 48 h, the global protein synthesis rate was assessed using the Surface Sensing of Translation (SUnSET) method [2,12,13]. Briefly, the cells were incubated with 4 µM puromycin dihydrochloride (Sigma-Aldrich, Milan, Italy, Cat. P8833) for 3 h, and were lysated with Laemmli Sample Buffer 4X (Bio-Rad, Hercules, CA, USA), and prepared for WB analysis.

##### Western Blot

Cell lysates were boiled at 96°C for 5 min. Then samples were loaded on a 12% polyacrylamide-sodium dodecyl sulphate gel for SDS-PAGE (sodium dodecyl sulphate—polyacrylamide gel electrophoresis). Proteins were transferred onto nitrocellulose membrane, using Mini Transfer Packs, with Trans-Blot^®^ Turbo TM (Bio-Rad) according to manufacturer’s instructions (Bio-Rad). Ponceau staining was acquired with the ChemiDocTM Imaging System (Bio-Rad), and was used to normalize protein loading. The membranes were blocked in 5% skim milk (Sigma, Cat. 70166) for 45 min at room temperature. Subsequently, membranes were incubated with anti-puromycin (Millipore, Burlington, MA, USA, Cat. MABE343) primary antibody, overnight at 4 °C. 

Goat anti-mouse IgG (H+L) horseradish peroxidase-conjugated secondary antibody (Bio-Rad, 1:5000; Cat. 170-6516,) was used as secondary antibodies. Detection was carried out with SuperSignalTM West Pico Chemiluminescent Substrate (Thermo Scientific), based on the chemiluminescence of luminol and developed using ChemiDocTM Imaging System (Bio-Rad).

#### 3.7.2. Melanoma Cell Lines Experiments

##### Real-Time PCR

Melanoma cell lines were plated in 6-well plates, as described in Section 3.5.2. After 24 h, cells were exposed (for 24 h) to TG, in the presence or absence of 4-PBA (5 µM; as described in Section 3.6.3), and Trizol reagent (Invitrogen, Burlington, ON, Canada) was used to isolate total RNA, as indicated by the supplier. Then, 2 µg of total RNA were used to generate cDNA by using the ExcelRT Reverse Transcriptase kit (Smobio, Rome, Italy), following the manufacturer’s recommendations. Quantitative PCR reactions were performed by using the Excel-Tag FAST qPCR SybrGreen kit (Smobio) and the CFX96 thermocycler (Bio-Rad Laboratories, CA, USA). The primers sequence for all amplicons, designed by using the online IDT PrimerQuest Tool software (IDT, Integrated DNA Technologies Inc., IA, USA) are reported below: L34 forward: 5′-GTCCCGAACCCCTGGTAATAGA-3′L34 reverse: 5′-GGCCCTGCTGACATGTTTCTT-3′ATF4 forward: 5′-GTGGCCAAGCACTTCAAACC-3′ATF4 reverse: 5′-CCCGGAGAAGGCATCCTC-3′ATF6 forward: 5′-TATCAGTTTACAACCTGCACCCACTA-3′ATF6 reverse: 5′-GCAAGGACTGGCTGAGCAGA-3′

Results were normalized by using human L34 as internal control, and comparative Ct method (ΔΔCt) was used for the relative quantification of gene expression [8].

#### 3.7.3. Statistical Analysis

The experiments were performed in triplicate and repeated three times. The ANOVA statistical analysis was carried out using GraphPad software (GraphPad Prism 6) by one- way ANOVA with Sidak’s multiple comparison (* *p* < 0.05; ** *p* < 0.01; *** *p* < 0.001; **** *p* < 0.0001). Concerning the experiments conducted in WT- and 3xTg-iAstro cells, the statistical differences were calculated by setting the astrocyte conditioned medium (ACM) as the control (Figure 2), whereas the expression of ER stress markers ATF4 and ATF6 in melanoma cell lines along the experimental conditions was evaluated by using DMEM blank and DMEM+4-PBA (5 µM) as negative controls (Figure 3). Finally, quantified 4-PBA concentrations after treatment were compared with the cell culture media including 4-PBA (NBA, 3 or DMEM, 5 µM) not exposed to cells (Figure 2 and Figure 3).

#### 3.7.4. Adsorption Evaluation of 4-PBA in Cell-Culture Dish

Adsorption assays were conducted in quadruplicate using initial concentrations of 3.00 or 5.00 µM in NBA and DMEM, which were moved into cell-culture plates without cells to observe the adsorption phenomenon singularly. After that, multiple aliquots were collected over 24 or 48 h (0 h; 1 h; 2 h, 4 h, 8 h, 24 h, and 48 h) that were the final incubation times in DMEM and NBA experiments, in order to plot an adsorption curve as a function of time. The “log (inhibitor) vs response (three parameters)” model of GraphPad Prism version 6 was used to represent graphs. The quantified concentrations were considered acceptable with a %RSD within ±15% (Relative Standard Deviation). Adsorption preliminary assays of 4-PBA employed an initial concentration of 3.00 µM in NBA and were performed on Pyrex glass petri dish, non-treated polystyrene petri dish (Corning; Cat. B51026), and plasma-treated polystyrene plasticware (Corning; Cat. 353046) used in this work, monitoring its free concentration at the outset (t0) and after 48 h of incubation (t48) by LC-HRMS. 4-PBA adsorbed fractions were calculated by subtracting 4-PBA initial concentration at t0 and final concentrations. Adsorption percentages were calculated by dividing the adsorbed fraction with the initial concentration of 4-PBA multiplied by 100.

## 4. Conclusions

In this work, a new LC-HRMS method for the quantification of 4-PBA was developed and successfully validated in two media, commonly used for in vitro cell cultures: NBA (NeuroBasal-A medium) and DMEM (Dulbecco’s Modified Eagle Medium). Our results suggested that the free concentration of 4-PBA was affected by adsorption during in vitro incubation. Firstly, upon addition of a nominal concentration, within an hour, ∼40% of 4-PBA is held by matrix components, such as ALB. In addition, a further loss of about 10% occurs during 24 h in DMEM cultures of melanoma cells, similarly in NBA for immortalized astrocytes after 48 h, reaching an overall loss of about 50%. Nevertheless, the remaining concentration is sufficient to produce an effect on the protein synthesis of WT-iAstro but not 3Tg-iAstro cells. In addition, we show that TG, which is a popular ER stress/UPR-inducing drug, does not affect the absorption kinetics of 4-PBA. However, its intracellular concentration normalized TG-induced ER stress in both A375 and CHL-1 cells.

Altogether, our paper demonstrates that the actual concentration of 4-PBA in a culture medium depends on matrix composition and the duration of the treatment, highlighting the importance of the control over 4-PBA in biological and drug discovery fields. In conclusion, our LC-HRMS method provides a versatile platform for the control of 4-PBA concentrations in culture media and paves the way for 4-PBA detection in other biological fluids.

## Figures and Tables

**Figure 1 pharmaceuticals-16-00298-f001:**
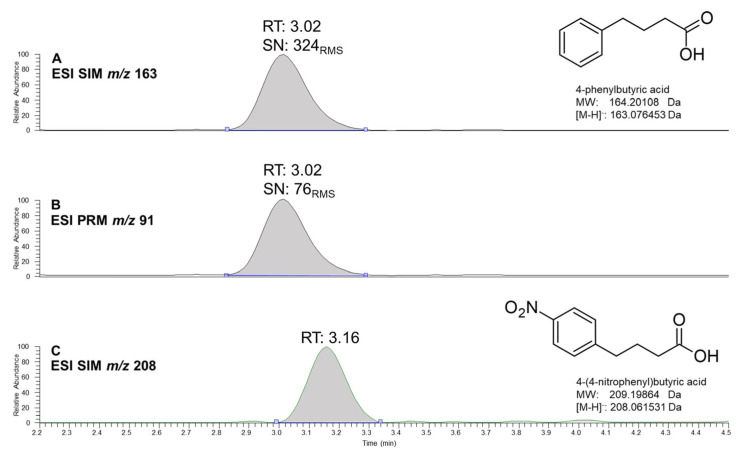
Chemical structures and chromatograms of 4-phenylbutyric acid (4-PBA) and 4-(4-nitrophenyl)butyric acid (NPBA, IS). (**A,B**) allow to compare the signal to noise ratio of 4-PBA in SIM and PRM acquisition modes, respectively. (**C**) shows IS (NPBA) chromatogram recorded in SIM.

**Figure 2 pharmaceuticals-16-00298-f002:**
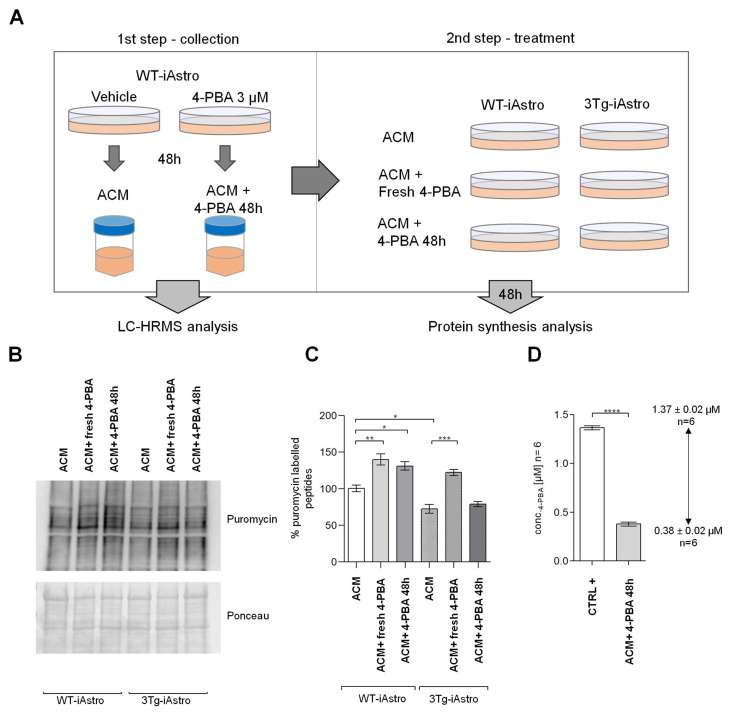
Treatment of Alzheimer’s astrocytes at various 4-PBA concentrations in conditioned medium. (**A**) WT and 3Tg-iAstro were treated with astrocyte conditioned medium (ACM), astrocyte conditioned medium with 4-PBA 3 µM freshly added (ACM + fresh 4-PBA), or astrocyte conditioned medium collected form astrocytes treated with 4-PBA for 48 h (ACM + 4-PBA 48 h). After 48 h, cells were pulsed with 4 µM puromycin and lysate for 1 h, then protein synthesis rate was accessed via WB. Concentrations of 4-PBA were assessed by LC-HRMS. (**B**) Representative WB of protein synthesis analysis, (**C**) data are expressed as mean ± SEM of 3 independent experiments, (**D**) 4-PBA absorption determined by LC-HRMS; concentrations are expressed as µM ± SEM of 3 independent experiments repeated twice. (** p* < 0.05; *** p* < 0.01; **** p* < 0.001*; **** p* < 0.0001; by one-way ANOVA, Sidak’s multiple comparison.).

**Figure 3 pharmaceuticals-16-00298-f003:**
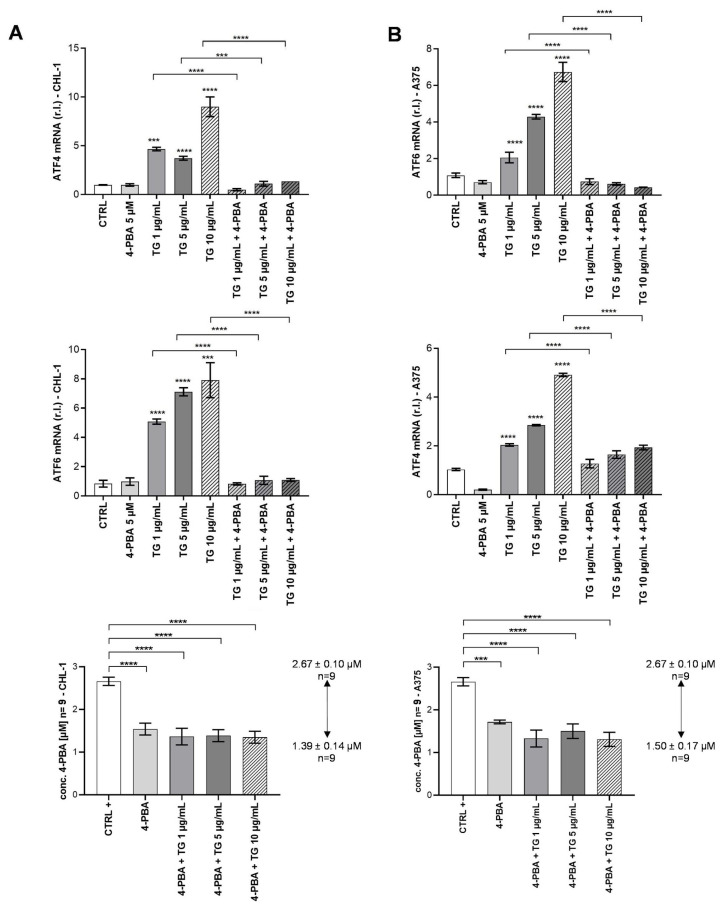
Thapsigargin-induced ER stress markers monitoring related to 4-PBA absorption from DMEM. (**A**,**B**) UPR analysis following TG and 4-PBA treatment: ATF4 and ATF6 gene expression was evaluated in CHL-1 and A375 cell lines after 24 h, by qPCR analysis. The expression of the ribosomal factor L34 was used as an internal control. Absorption data were determined in CHL-1 and A375 cell lines after 24 h, by LC-HRMS (right histograms). CTRL+ represents medium + 4-PBA not exposed to cells. 4-PBA concentrations are expressed as µM ± SE. Experiments were performed in triplicate and repeated three times. The histograms represent the mean ± SD; *** *p* < 0.001; **** *p* < 0.0001.

**Figure 4 pharmaceuticals-16-00298-f004:**
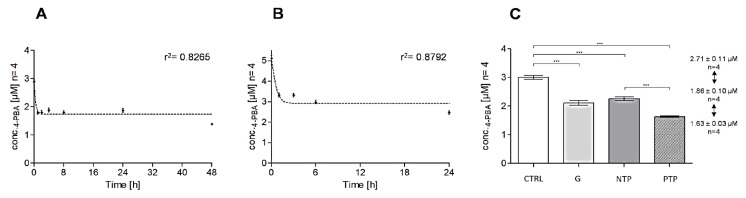
Adsorption kinetics of 4-PBA in cell culture plates. (**A**,**B**) shows the adsorption curves as a function of time, respectively in NBA and DMEM. X axes represent time expressed in hours [h] and Y axes the concentration of free 4-PBA in solution [µM]. (**C**) Glass petri (G), non-treated polystyrene petri dishes (NTP) and plasma-treated polystyrene plasticware (PTP) showed a similar adsorption with a decrease of about 40%, which is in line with the adsorption curves (**A**,**B**). (*** *p* < 0.001; by one-way ANOVA, Sidak’s multiple comparison).

**Table 1 pharmaceuticals-16-00298-t001:** Validation parameters in NBA matrix. For each QC level, precision intra and inter-day expressed as %RSD, accuracy (AC; w.r., within-run; b.r., between-run) within and between-run, matrix effect (ME), and recovery are reported.

Linearity	Y = 10.7600 (±0.0524)∙x + 0.5943 (±0.1717) *				*r*^2^ = 0.9993	
QC Level	Intra-Day (%RSD)	Inter-Day (%RSD)	AC (w.r.)	AC (b.r.)	ME	Recovery
	n = 5	n = 15	n = 5	n = 15	n = 3	n = 3
LLOQ	8.80	10.68	1.74	−0.63	-	-
LQC	7.86	9.16	4.69	0.32	114.0	114.2
MQC	1.67	6.98	1.40	−1.68	98.8	111.1
HQC	1.95	5.99	−0.04	−2.84	97.9	111.3

* Bracketed values indicate SEs of the regression slope and intercept.

**Table 2 pharmaceuticals-16-00298-t002:** Validation parameters in DMEM matrix. For each QC level, precision intra and inter-day expressed as %RSD, accuracy (AC; w.r., within-run; b.r., between-run) within and between-run, matrix effect (ME), and recovery are reported.

Linearity	Y = 7.8300 (±0.1121)∙x + 0.7300 (±0.3671) *				*r*^2^ = 0.9939	
QC Level	Intra-Day (%RSD)	Inter-Day (%RSD)	AC (w.r.)	AC (b.r.)	ME	Recovery
	n = 5	n = 15	n = 5	n = 15	n = 3	n = 3
LLOQ	8.53	9.29	−10.31	−7.42	-	-
LQC	14.65	12.45	7.11	13.89	85.6	92.5
MQC	6.69	6.38	4.16	5.53	65.0	98.6
HQC	7.41	8.59	1.01	3.74	61.4	95.3

* Bracketed values indicate SEs of the regression slope and intercept.

**Table 3 pharmaceuticals-16-00298-t003:** Stability assays in NBA matrix. Stability assays were carried out at LQC, MQC, and HQC level, precision expressed as %RSD and accuracy (AC; w.r., within-run; b.r., between-run) are reported for each of them.

	Autosampler Stability 48 h		Bench Top Stability 6 h		Freeze–Thaw Cycles (I, II, III)		Stability after Solvent Evaporation	
QC Level	Area (%RSD)	AC (w.r.)	Area (%RSD)	AC (w.r.)	Area (%RSD)	AC (b.r.)	Area (%RSD)	AC (w.r.)
	n = 3	n = 3	n = 3	n = 3	n = 9	n = 9	n = 3	n = 3
LQC	0.92	6.39	6.93	−12.73	9.79	−0.86	12.42	−13.42
MQC	2.69	−1.56	2.44	−9.95	6.22	−1.55	2.14	−8.80
HQC	0.54	−1.61	0.29	−10.74	5.32	−1.45	1.62	−10.28

**Table 4 pharmaceuticals-16-00298-t004:** Stability assays in DMEM matrix. Stability assays were carried out at LQC, MQC, and HQC level, precision expressed as %RSD and accuracy (AC; w.r., within-run; b.r., between-run) are reported for each of them.

	Autosampler Stability 48 h		Bench Top Stability 6 h		Freeze–Thaw Cycles (I, II, III)		Stability after Solvent Evaporation	
QC Level	Area (%RSD)	AC (w.r.)	Area (%RSD)	AC (w.r.)	Area (%RSD)	AC (b.r.)	Area (%RSD)	AC (w.r.)
	n = 3	n = 3	n = 3	n = 3	n = 9	n = 9	n = 3	n = 3
LQC	5.26	15.00	10.64	3.11	11.52	11.94	13.89	2.90
MQC	1.68	−5.93	4.15	1.14	9.44	12.82	2.88	6.45
HQC	7.94	−8.46	5.30	4.13	11.83	11.30	0.20	7.21

## Data Availability

Data is contained within the article and Appendix A.

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
