# Peer review of "Quantification of the Chemical Chaperone 4-Phenylbutyric Acid (4-PBA) in Cell Culture Media via LC-HRMS: Applications in Fields of Neurodegeneration and Cancer"

_pharmaceuticals, 2023, doi:10.3390/ph16020298_

Round 1

Reviewer 1 Report

The submitted manuscript lacks novelty. The determination of phenylbutyric acid and its metabolite phenylacetic acid was studied previously in 2012 doi: 10.1016/j.jchromb.2012.07.004. line 317. To the best of our knowledge, our method represents the first validated system...assessment of 4-PBA in NBA (NeuroBasal-A medium). ere is in the following link is expressing the  4-Phenylbutyrate ameliorates apoptotic neural cell death: doi: 10.1038/s41598-020-70362-x.

lines  172-173: Generation of immortalized astrocytes from hippocampi of WT and 3xTg-AD mice (WT- and 3Tg-iAstro cells) was described elsewhere.27 

The word elsewhere is not scientific. 

Author Response

Dear Reviewer,

we wish to thank you for your revision. Here enclosed you can find the answers to your indications:

Question 1: The submitted manuscript lacks novelty. The determination of phenylbutyric acid and its metabolite phenylacetic acid was studied previously in 2012 doi: 10.1016/j.jchromb.2012.07.004. line 317. To the best of our knowledge, our method represents the first validated system...assessment of 4-PBA in NBA (NeuroBasal-A medium). ere is in the following link is expressing the 4-Phenylbutyrate ameliorates apoptotic neural cell death: doi: 10.1038/s41598-020-70362-x.

Answer 1: Thanks for the feedback on the novelty; we only want to underline that, even though apparently there is a similarity with the paper you mentioned and that we have already and deeply read, this is not really true. In fact, in an analytical point of view, we performed a validation in diverse matrices, which are cell culture media, different from plasma. The applications are divergent too, because we have worked in vitro whereas the mentioned colleagues in vivo. The second article highlighted only biological results obtained by Western Blot or quantitative PCR to consider the expression levels of specific markers, it does not describe any liquid chromatography-mass spectrometry method. Furthermore, there is not a direct correlation between the free concentration of 4-PBA and the observed effects. Furthermore, they treated their cells with a higher concentration of 4-PBA (100 µM), whereas we found appreciable rescue effects even at 3.0-5.0 µM.

Question 2: lines 172-173: Generation of immortalized astrocytes from hippocampi of WT and 3xTg-AD mice (WT- and 3Tg-iAstro cells) was described elsewhere.27…. The word elsewhere is not scientific.

Answer 2: We apologize for the lacking information, and as you suggested we implemented it in the text by giving more details about the followed procedures and/or the origin of our cells, as you may observe below:

2.5.1 Immortalized hippocampal astrocytes from WT and 3xTg-AD mice

Immortalized astrocytes from hippocampi of WT and 3xTg-AD mice (WT- and 3Tg-iAstro cells) were generated in our laboratory via transduction using retrovirus expressing SV40 large T antigen. The procedural details are available in the following reference. [27]

2.5.2 Melanoma models A375 and CHL-1 cell lines

Melanoma cell lines, CHL-1 (Cellosaurus Resource Identification Initiative, RRID: CVCL_1122) and A375 (Cellosaurus RRID: CVCL_0132) were a gift from Prof. P.E. Lovat (Newcastle University, Newcastle Upon Tyne, UK).

Reviewer 2 Report

The manuscript pharmaceuticals-2084776 is very interesting, within the journal scope, and can be important for several researchers in the same field. A few points must be clarified before its publication. The authors must explain in more detail the validation method; for example, in Supporting Information, because for some researchers, access to the bibliography given can be difficult. The statistical point (2.7.2.2 Statistical analysis ) should be more detailed. The authors should carefully explain the type of control used in each experiment. Several abbreviations are used without being defined. Although they are evident for researchers working in the field, the authors should be aware that their manuscript is also for other researchers less acquainted with those abbreviations.

Author Response

Reviewer 2.

Dear Reviewer,

we wish to thank you for your revision. Here enclosed you can find the answers to your indications.

Question 1: The manuscript pharmaceuticals-2084776 is very interesting, within the journal scope, and can be important for several researchers in the same field. A few points must be clarified before its publication. The authors must explain in more detail the validation method; for example, in Supporting Information, because for some researchers, access to the bibliography given can be difficult.

Answer 1: Thank you very much for your suggestions. We reported the information you requested about validation method as supplemental material, directly available from the official resources as pdf.

Question 2: The statistical point (2.7.2.2 Statistical analysis ) should be more detailed. The authors should carefully explain the type of control used in each experiment.

Answer 2: We apologize for the lacking details. We followed your suggestions by supplementing the ex paragraph “2.7.2.2 Statistical analysis” with more information for each experimental set, we recalled it “2.7.3 Statistical analysis”. We also clarified the type of control used in each experiment both in the text and in captions, as you may read below:

“2.7.3 Statistical analysis

The experiments were performed in triplicate and repeated three times. The ANOVA statistical analysis was carried out using GraphPad software (GraphPad Prism 6) by one- way Anova with Sidak’s multiple comparison (* p <0.05; ** p <0.01; *** p <0.001; **** p <0.0001). Concerning the experiments conducted in WT and 3xTg-iAstro cells, the statistical differences were calculated by setting the astrocyte conditioned medium (ACM) as control (Figure 2). Whereas, the expression of ER stress markers ATF4 and ATF6 in melanoma cell lines along the experimental conditions was evaluated by using DMEM blank and DMEM+4-PBA (5 µM) as negative controls. Finally, quantified 4-PBA concentrations after treatment were compared with the cell culture media including 4-PBA (NBA, 3 or DMEM, 5 µM) not exposed to cells (Figures 2-3).”

Question 3: Several abbreviations are used without being defined. Although they are evident for researchers working in the field, the authors should be aware that their manuscript is also for other researchers less acquainted with those abbreviations.

Answer 3: Thank you for highlighting this trouble. We have resolved by including the meaning of all used abbreviations and acronyms in the text, as listed below:

RSDs (Relative Standard Deviations) - abstract

3xTg-AD mice (Triple-Transgenic model (3xTg-AD) harboring PS1(M146V), APP(Swe), and tau(P301L) transgenes) - introduction

CNS (Central Nervous System) - introduction

ATF (Activating Transcription Factors) - introduction

LLOQ (Lower Limit of Quantification) - 2.2 Preparation of standard solutions

Quality Controls (QCs) - 2.2 Preparation of standard solutions

Immortalized astrocytes (iAstro) - 2.5.1 Immortalized hippocampal astrocytes from WT and 3xTg-AD mice

WT (Wild Type) - 2.5.1 Immortalized hippocampal astrocytes from WT and 3xTg-AD mice

SDS-PAGE (Sodium Dodecyl Sulphate - PolyAcrylamide Gel Electrophoresis) - 2.7.1.2 Western Blot

Reviewer 3 Report

In this paper, the authors reported and validated a new LC-HRMS method for the quantification of 4-PBA in two media for in vitro cell cultures (NBA 458 and DME). They demonstrated that free concentration of 4-PBA was affected by adsorption during in vitro incubation, affecting the protein synthesis of different cells. The data are well presented and commented, supporting the conclusion that the method can be applied for the quantification of 4-PBA.

I suggest to change some minor points:

- The quality of Figure 2 and 3 is poor, the letters and numbers are too small to read.

- Figure 2C is not well commented both in the text and in the figure caption

- The data reported in supplementary materials are not cited and commented in the text

Author Response

Reviewer 3.

Dear Reviewer,

we wish to thank you for your revision. Here enclosed you can find the answers to your indications.

Question 1: The quality of Figure 2 and 3 is poor, the letters and numbers are too small to read.

Answer 1: Thank you very much for your advice. Actually, it was helpful to increase the quality of our paper. Herein, we increased both the quality and the size of Figures 2-3, in order to facilitate their examination, as you can see in the text.

Question 2 and 3: - Figure 2C is not well commented both in the text and in the figure caption

- The data reported in supplementary materials are not cited and commented in the text”

Answer 2: We improved the connection between the text and the given results, plus the supplementary information, in which we proposed a new table including all the settings of our liquid chromatography-mass spectrometry method. Below you may consider the changes we brought:

3.2 4-PBA quantification in astrocyte-conditioned Neurobasal A medium

[...] ACM+ 4-PBA 48h, in which a concentration of 4-PBA 0.38 ± 0.02 µM was present in the medium, was still able to increase protein synthesis in WT-iAstro but had no effect on 3Tg-iAstro cells, suggesting that higher 4-PBA concentrations are necessary to rescue protein synthesis in 3Tg-iAstro (Figure 2C). [...]

Figure 2. [... ] (B) representative WB of protein synthesis analysis, (C) which data are expressed as mean ± SEM of 3 independent experiments [...]

[...] thapsigargin (TG) ([M-H]-: 649.32295 m/z), which chromatograms are reported in Supplementary Material (Figure 1S). MS spectra were acquired and processed using Xcalibur® software (Thermo Fisher Scientific). A summary table of our LC-HRMS method is available in the Supplementary Material (Table 1S). - 2.4.1 LC-HRMS analyses

[...] Chemical structures and chromatograms of both analyte and IS are shown in Figure 1, whereas MS/MS (Tandem mass spectrometry) spectra used for their structural confirmation are reported in the Supplementary Material (Figures 2S, A-C). - 3.1 Development and optimization of LC-HRMS conditions and sample preparation

[...] Hence, we were able to semi-quantify thapsigargin, even if it were not our main aim we confirmed its operating concentrations (Supplementary material - Figure 3S). - 3.1 Development and optimization of LC-HRMS conditions and sample preparation

Reviewer 4 Report

Minor remarks

Lines 107-108: ”where 100 μL of IS solution 0.02 107 μM IS solution”. This part of the sentence is not clear. Please explain this to make it clear.

All other minor remarks are depicted in the document.

 Major remarks

There are a lot of cited references. Please, consider reducing the references list. 

Author Response

Reviewer 4.

Dear Reviewer,

Before giving you a rundown of the modifications applied to our manuscript, we wish to thank you for your revision. We updated our paper, on the base of your suggestions.

Question 1: There are a lot of cited references. Please, consider reducing the references list.

Answer 1: We took into consideration your advice by reducing the references, specifically we deleted the following ones which were truely redundant:

  1. DÄ…browski, A. Adsorption — from Theory to Practice. Adv. Colloid Interface Sci. 2001, 93 (1–3), 135–224. https://doi.org/10.1016/S0001-8686(00)00082-8.
  2. Zheng, X.; Baker, H.; Hancock, W. S.; Fawaz, F.; McCaman, M.; Pungor, E. Proteomic Analysis for the Assessment of Different 604 Lots of Fetal Bovine Serum as a Raw Material for Cell Culture. Part IV. Application of Proteomics to the Manufacture of Biolog-605 ical Drugs. Biotechnol. Prog. 2008, 22 (5), 1294–1300. https://doi.org/10.1021/bp060121o.

Question 2: Lines 107-108: where 100 μL of IS solution 0.02 107 μM IS solution. This part of the sentence is not clear. Please explain this to make it clear.

Answer 2: Thank you for highlighting this typing mistake. You may read the corrected new version below, we also added more details:

“2.2 Preparation of standard solutions

[...] IS stock and its diluted solutions were prepared using methanol as solvent. IS stock 20 mM solution was diluted to reach the final concentration of 0.02 µM and used to precipitate matrix proteins during sample preparation. Blank and zero samples were made using free-analyte matrices. In the first case, IS was not added either. Matrix-based standards (calibrators, QCs, blank and zero samples) were freshly prepared.”

We are also grateful to you, because your review report with the remarks helped us to speed up the revision process; we considered all your indications and implemented them in the revised manuscript.